# Corifolitropin-Alfa plus Five Days Letrozole Versus Daily Recombinant-FSH in Expected Normo-Responder Patients: A Retrospective Comparative Study

**DOI:** 10.3390/diagnostics13071249

**Published:** 2023-03-27

**Authors:** Giuseppe D’Amato, Anna Maria Caringella, Antonio Stanziano, Clementina Cantatore, Antonio D’Amato, Ettore Cicinelli, Amerigo Vitagliano

**Affiliations:** 1Department of Advanced Reproductive Risk Management and High-Risk Pregnancies, ASL Bari, Reproductive and IVF Unit, PTA “F Jaia”, 70014 Conversano, BA, Italy; 2Unit of Obstetrics and Gynecology, Department of Interdisciplinary Medicine (DIM), University of Bari, 70100 Bari, BA, Italy

**Keywords:** infertility, ovarian stimulation, expected normal ovarian responder, letrozole, corifolitropin-α, recombinant-FSH

## Abstract

**Background:** In recent times, different novel GnRH-antagonist protocols with various combinations of gonadotropins and other molecules (e.g., aromatase inhibitors, selective estrogen receptor modulators) have been proposed for expected normal ovarian responders undergoing assisted reproductive treatments. The purpose of this study was to evaluate the effectiveness of a novel ovarian stimulation protocol based on the combination of corifollitropin-alfa plus five days of letrozole in E-NOR women undergoing IVF as compared with a daily recombinant-FSH regimen. **Methods:** We conducted a retrospective-controlled study on 182 couples undergoing their first IVF attempt. In Group A (experimental), letrozole (2.5 mg daily) was administered from day 2 (up to day 6 of the cycle), followed by corifollitropin-alfa on day 3 and daily recombinant FSH from day 10. In Group B, recombinant FSH from day 2 were administered (150 IU-225 IU daily). Statistical analysis was completed using SPSS Statistics. The primary outcome was the total number of MII oocytes retrieved. **Results**: Group A showed similar results compared to Group B in terms of MII oocytes, live birth, implantation, and clinical pregnancy rates (*p* = ns). Nevertheless, the experimental group was associated with a trend towards a higher number of developing follicles, total oocytes, and embryos (*p* < 0.05) with lower estradiol and progesterone values at ovulation induction compared to Group B, resulting in an increased chance of performing a fresh embryo transfer (*p* < 0.05). **Conclusions**: The combination of CFα plus five days of letrozole was associated with a trend towards a higher number of developing follicles, total oocytes, and obtained embryos. Moreover, the experimental protocol resulted in lower estradiol and progesterone values at ovulation induction compared to daily rFSH, with an increased chance of performing a fresh embryo transfer (with no OHSS occurrence). Given the observational design of our study, further well-conducted RCTs are needed.

## 1. Introduction

As the demand for IVF treatments continues to increase globally, the need for safe and effective ovarian stimulation protocols has become an essential aspect of reproductive medicine [1,2]. E-NOR patients, who account for approximately one-third of IVF cases, are a crucial group in this regard. E-NOR patients are typically women of intermediate reproductive age who have undergone normal ovarian reserve testing, are expected to have adequate oocyte recovery, and are at a low risk of developing ovarian hyperstimulation syndrome (OHSS) [3,4]. As such, finding optimal ovarian stimulation protocols for this group is of paramount importance in IVF treatment. The current strategy for ovarian stimulation in E-NOR includes the administration of daily gonadotropins at a fixed dose in combination with GnRH-antagonists for the prevention of premature LH rise. This approach is increasingly taking over the “old” long GnRH-agonist protocol for safety reasons due to the possibility of triggering ovulation with GnRH-agonists in cases of increased risk of OHSS [4,5,6]. However, also this approach is not free from limits, including several daily injections and increased risk in case of cycle segmentation (i.e., additional costs related to embryo freezing and thawing, gynecological consultations, and drugs).

In recent times, different novel GnRH-antagonist protocols, with various combinations of gonadotropins and other molecules (e.g., aromatase inhibitors, selective estrogen receptor modulators), have been proposed for E-NOR [7,8]. The reason for creating new protocols in IVF programs is to make them more affordable and efficient and to reduce the amount of stress and difficulty involved in undergoing treatment.

Corifollitrophin-alfa (CFα) is a long-acting follicle-stimulating hormone analog that was developed by fusing the C-terminal peptide of the β-subunit of human chorionic gonadotropin (hCG) to human follicle-stimulating hormone (FSH), resulting in a single hybrid molecule. This fusion protein extends the half-life of FSH, allowing for sustained stimulation of follicular growth and maturation, thus reducing the frequency of injections required during assisted reproductive technology (ART) cycles (i.e., potentially replacing seven daily injections of recombinant FSH [rFSH]) [9]. A recent meta-analysis of randomized controlled trials showed that CFα was not inferior to daily rFSH injections in E-NOR patients undergoing IVF/ICSI treatment cycles, leading to similar success rates and risks of OHSS [10].

Letrozole, a third-generation non-steroidal aromatase inhibitor, is a versatile drug that has found use in several different contexts. Its most common application is as an adjunct treatment for breast cancer patients undergoing oocyte cryopreservation cycles. In this setting, letrozole is used to suppress plasma estrogen levels during controlled ovarian stimulation, which is an essential component of fertility preservation in breast cancer patients [11,12]. However, letrozole has also demonstrated efficacy in stimulating multifollicular growth through a variety of mechanisms. One of the key mechanisms by which it achieves this effect is by competitively binding to the aromatase enzyme, thereby inhibiting the conversion of androstenedione and testosterone to estrone and estradiol. This results in increased intrafollicular concentrations of androgens, which can stimulate the proliferation of granulosa and theca cells, promote the growth of small follicles, and upregulate the expressions of FSH, IGF-I, and IGF-I receptors [11,12,13,14,15]. Moreover, letrozole has been shown to increase the central release of FSH by reducing the negative feedback effect of estradiol production by the ovary. This effect may facilitate early follicle recruitment and reduce the required amount of exogenous FSH during controlled ovarian stimulation (COS) [15,16]. Due to these mechanisms, letrozole has been proposed as a co-treatment during ovarian stimulation for in vitro fertilization (IVF) cycles, although the results of studies investigating its effectiveness in this context have been somewhat controversial [13,14].

To date, no study has tested the combination of CFα plus letrozole in E-NOR patients undergoing IVF. Herein, our purpose was to evaluate the effectiveness of a novel ovarian stimulation protocol based on the combination of CFα plus five days of letrozole in E-NOR women undergoing IVF as compared with a daily rFSH regimen.

## 2. Materials and Methods

### 2.1. Study Design

This was a retrospective controlled study conducted at a public IVF center (“Center of IVF and Human pathophysiology,” Conversano, Bari, Italy) from January 2019 to December 2021. The study was approved by the local ethics committee (n° 5850). All patients gave their written consent to using their data for research purposes.

### 2.2. Patients

During the study period, 182 consecutive couples were included. Inclusion criteria were: female age ≤38 years; anti-mullerian hormone (AMH) levels ≥1.5 ng/mL and <4 ng/mL; female body mass index <32 kg/m^2^; normal karyotype (for both partners); and a first IVF attempt.

Exclusion criteria were: previous IVF attempts using the COS protocols applied in this study; a history of recurrent miscarriage (i.e., ≥2 consecutive spontaneous abortions); intrauterine pathologies, myomas with uterine cavity distortion, autoimmune diseases, and a history of oncologic diseases. Azoospermic men requiring surgical sperm extraction were also excluded from the study.

In this study, the allocation of patients to one group or another took place over two different time periods. From August 2020 to December 2021, all the patients matching our inclusion criteria received the “experimental protocol” (*n* = 91; Group_A). These patients were retrospectively matched in a 1 to 1 ratio with consecutive patients treated with rFSH from July 2020 backwards (up to January 2019; *n* = 91; Group_B).

### 2.3. M-COH Protocols

COS was started on the second day of the spontaneous menstrual cycle in both treatment harms. In Group A, oral letrozole (2.5 mg daily) was administered from day 2 (up to day 6 of the cycle), followed by a single injection of CFα on day 3 (100 or 150 mcg following European Medical Agency indications). From day 10, daily rFSH was administered, when necessary, until ovulation induction (150UI-225UI/day).

In Group B, ovarian stimulation was completed by using daily injections of rFSH from day 2 until ovulation induction at a starting dose of 150 IU-225 IU per day based on the Nelson et al. model [17]. Based on the ovarian response, as assessed by ultrasound examination and serum hormonal measurements (including follicular stimulating hormone, FSH; luteinizing hormone, LH; estradiol, E2; and progesterone, P) every two days, the dose of gonadotropins was adjusted, if needed.

The first monitoring ultrasound scan and E2 assay were performed on day 5 of the cycle, and the administration of GnRH antagonist was started when the leading follicle size was ≥14 mm. When two leading follicles were ≥17–18 mm diameter, ovulation was inducted by using human chorionic gonadotropin (hCG) at 10.000 IU s.c. (Gonasi©, IBSA, Lodi, Italy), followed by oocyte retrieval 35–36 h later. In cases of high ovarian response (i.e., ≥15 follicles ≥12 mm on the day of the trigger) or premature luteinization (i.e., serum progesterone >1.5 ng/mL), ovulation induction was carried out by administering triptorelin 0.2 mg.

### 2.4. Oocyte Insemination and Embryo Transfer

Oocytes were inseminated by IVF or intracytoplasmic sperm injection (ICSI) according to sperm parameters. Fertilization was checked by the embryologists one day after IVF/ICSI, and the embryos were cultured until the blastocyst stage (days 5–6), preferably. When ≤3 viable embryos were identified on culture day 3, the culture was stopped and the embryos were transferred and/or cryopreserved. For those women in whom ovulation was induced with GnRH-agonists, all the embryos were cryopreserved. Embryo cryopreservation was performed using the vitrification technique. Vitrification/warming were performed by two operators using the kit produced by Kitazato BioPharma Co., (Fuji, Japan).

When the freeze all strategy was adopted, the embryo transfers were done using conventional hormonal replacement therapy.

The number of embryos to be transferred was chosen based on the woman’s age, embryonic stage, and quality (according to Gartner criteria). In women ≤35 years old, only single embryo transfers were offered. For women >35 years old, the number of transferred embryos was based on embryonic stage and quality. The luteal phase was supported with vaginal progesterone capsules (600 mg daily), starting from the day of the oocyte retrieval (or based on the embryonic stage in frozen-thawed embryo transfer cycles), for 14 days. Serum hcg measurement was performed 14 days after the oocyte retrieval. In cases of positive results, progesterone administration continued until the seventh gestational week.

### 2.5. Data Collection and Reproductive Follow-Up

Data was retrieved from the patient’s clinical charts. Information on the obstetrical outcomes was obtained through telephone interviews. In the case of a positive pregnancy test, patients were followed up until delivery (in the case of a live birth).

The primary outcome was the total number of mature (MII) oocytes retrieved. Secondary outcomes were: total oocytes retrieved, clinical pregnancy rate (CPR), implantation rate (IR), miscarriage rate (MR), and live birth rate (LBR). Tertiary outcomes were: total units of rFSH, serum estradiol and progesterone at ovulation induction, number of daily antagonist injections, and rate of fresh embryo transfers.

### 2.6. Outcomes Measures

Mature oocytes were the total number of MII oocytes after decumulation. Total oocytes were the total number of oocytes retrieved (including MII oocytes, MI oocytes, and germinal vesicles).

A live birth was defined as the birth of one or more living infants.

A clinical pregnancy was defined by the identification of a gestational sac with a fetal heart beat (FHB) at trans-vaginal ultrasound. The implantation rate was the ratio between the number of gestational sacs with FHB and the number of transferred embryos. A miscarriage is a pregnancy loss occurring before the 20th gestational week [2].

### 2.7. Statistical Analysis

Statistical analysis was performed using SPSS Statistics (version 22). Data was presented as means ± standard deviation (SD) or as a number (percentage). Comparisons between categorical variables were made by using contingency tables and the chi-squared test or Fisher’s exact test, when needed. Comparisons between normally distributed continuous variables were performed using the student’s *t*-test. A value of *p* < 0.05 was considered statistically significant.

## 3. Results 

### 3.1. Baseline Characteristics

Among the 182 couples included in the study, *n* = 91 were included in Group_A and *n* = 91 in Group_B (Figure 1). The baseline characteristics of the two groups were comparable in all the variables (all *p* > 0.05), with the exception of LH (higher in Group_B; *p* = 0.001). There were no significant differences regarding the cause of infertility and the number of previous miscarriages (all *p* > 0.05). (Table 1)

### 3.2. Ovarian Stimulation Parameters

The duration of ovarian stimulation was similar between groups (11.39 ± 1.66 vs. 10.98 ± 1.37 [days]; *p* > 0.05). Group_A showed lower daily r-FSH consumption (692.7 ± 487.9 vs. 2396.5 ± 1122.1 [IU]; *p* < 0.0001), less GnRH-antagonist injections (4.06 ± 1.42 vs. 4.88 ± 1.24; *p* < 0.0001), a higher number of day 8 antral follicles (9.89 ± 4.4 vs. 5.74 ± 3.87; *p* < 0.0001), and a trend towards a higher number of pre-ovulatory follicles (8.74 ± 3.30 vs. 7.96 ± 2.61; *p* > 0.05), and a Group_B. Additionally, serum estradiol values at ovulation induction were lower in Group_A (1947.9 ± 1361.1 vs. 2681.2 ± 1569.2 [ng/mL]; *p* = 0.0009). Similarly, Group_A showed lower progesterone values at ovulation induction than Group_B (1.10 ± 0.57 vs. 1.35 ± 0.92 [ng/mL]; *p* = 0.03) (Table 2).

### 3.3. Oocyte Retrieval and Embryo Culture

Oocyte pick-up was completed without complications in both groups. Women in Group_A obtained a higher number of total oocytes (9.97 ± 3.81 vs. 8.76 ± 2.60; *p* = 0.01), with a borderline difference in terms of MII oocytes (7.98 ± 2.78 vs. 7.22 ± 2.47; *p* = 0.05) compared to Group_B (Figure 2). Fertilization rates were comparable between groups (0.79 ± 0.22 vs. 0.80 ± 0.19; *p* > 0.05). Notably, Group_A obtained a higher number of total embryos (3.73 ± 2.19 vs. 3.07 ± 1.22; *p* = 0.01), while the total number of good quality embryos/blastocysts were similar (2.72 ± 1.03 vs. 2.87 ± 1.19; *p* > 0.05) (Table 2).

### 3.4. Embryo Transfer Outcome

In Group_A, *n* = 67 women received a fresh embryo transfer versus *n* = 54 in Group_B (73.6% vs. 59.3%; *p* = 0.04). The number of transferred embryos was similar (1.52 ± 0.57 vs. 1.48 ± 0.61; *p* > 0.05), with no difference in good quality embryos/blastocysts (1.39 ± 0.62 vs. 1.41 ± 0.54; *p* > 0.05). Positive hcg tests at the first embryo transfer occurred in 49 and 42 women in Group_A and Group_B, respectively (53.8% vs. 46.1%, *p* > 0.05). IR were 38.4% and 35.5% (*p* > 0.05). No difference was detected in CPR (46.7% vs. 43.68%; *p* > 0.05), MR (16.67% vs. 11.76%; *p* > 0.05), and LBR (37.78% vs. 34.48%; *p* > 0.05). A single case of intrauterine fetal death occurred in Group_A.

A higher number of supernumerary embryos was obtained in Group_A (2.21 ± 1.04 vs. 1.58 ± 1.14 in Group_A and Group_B, respectively [*p* = 0.001]). LBR was not statistically different in women receiving fresh embryo transfer compared to those receiving frozen-thawed embryo transfer (35.61% vs. 38.83%, *p* > 0.05) (Table 2 and Figure 3).

## 4. Discussion

### 4.1. Main Findings

Although the primary outcome of the study, which aimed to investigate the effectiveness of a novel ovarian stimulation protocol based on the combination of CFα and letrozole in E-NOR patients undergoing IVF, did not show statistical differences between groups in terms of the number of MII oocytes obtained, the results did reveal some interesting findings. Specifically, the total number of retrieved oocytes was significantly higher in Group_A, suggesting a potential higher effectiveness of the experimental protocol compared to the daily r-FSH scheme. This finding aligns with the results of previous studies that have explored letrozole co-treatment during ovarian stimulation with gonadotropins in larger samples [8,14,18], leading us to speculate that the lack of a significant difference in MII oocytes between the groups may have been due to a Type II error. Furthermore, there was a noticeable trend towards a higher number of retrieved MII oocytes in Group_A compared to Group_B, even if statistical significance was not reached (7.98 ± 2.78 vs. 7.22 ± 2.47; *p* = 0.05), indicating the potential for further investigation of this ovarian stimulation protocol. Otherwise, if we look at the data from another perspective, the lower rate of oocyte maturity in Group_A was potentially attributable to an asynchrony between follicle growth and inner oocyte maturation due to the changes in follicular fluid dynamics exerted by letrozole. A previous study on letrozole use in women affected by breast cancer found a lower percentage of mature oocytes (73.2% for letrozole versus 86.3% for controls; *p* < 0.05) [19], but this occurrence was prevented by delaying administration of HCG until the larger follicle was 20 mm in mean diameter. Based on the findings of in vitro follicle culture by Hu et al. [20], this phenomenon is caused by the earlier development of the antral space in growing follicles exposed to aromatase inhibitors, with a delay in oocyte ripening compared to follicular growth. Notably, given that all the patients in our study were administered r-hcg when two dominant follicles reached 17–18 mm in diameter, we cannot exclude that postponing the administration of HCG might have increased the rates of MII oocytes in Group_A. On the other hand, we must stress that the findings by Oktay et al. [19] may not be fully applicable to our patients because of different duration of letrozole administration (i.e., entire duration of ovarian stimulation in Oktay et al. study versus initial five days in our experience).

In line with the increased number of total oocytes retrieved, Group_A showed a higher number of day 8 antral follicles (9.89 ± 4.4 vs. 5.74 ± 3.87; *p* < 0.0001) and a trend towards a higher number of pre-ovulatory follicles (8.74 ± 3.30 vs. 7.96 ± 2.61; *p* > 0.05) compared to Group_B. All these findings may suggest the existence of a synergistic action between letrozole and CFα in early follicle recruitment and the prevention of follicular atresia. On one hand, letrozole stimulates androgen production and Cyp17a1 mRNA expression [21], thus contributing to granulosa cell mitosis, sensitivity to FSH, and resistance to atresia. On the other hand, CFα exerts prolonged follicle-stimulating activity, initiating and supporting the growth of a large cohort of follicles during the first week of ovarian stimulation [22,23]. In this scenario, follicle recruitment is further boosted by the central release of endogenous FSH in response to hypoestrogenemia. All these biological mechanisms may explain the marked follicle growth we found in Group_A during the first half of ovarian stimulation, which was considerably higher compared to Group_B.

In addition to the aforementioned effects on folliculogenesis, the experimental protocol was associated with lower estradiolemia (1947.9 ± 1361.1 vs. 2681.2 ± 1569.2 [ng/mL]; *p* = 0.0009) and progesteronemia on the day of ovulation induction compared to controls (1.10 ± 0.57 vs. 1.35 ± 0.92 [ng/mL]; *p* = 0.03). While the reduction of estrogenemia is a well-described direct consequence of aromatase inhibition, our findings about lower progesterone values in Group_A require further discussion as they diverge from other authors’ findings. In this respect, a recent study failed to demonstrate an effect of letrozole in preventing premature luteinization in expected normo-responsive patients [14]. Globally, the occurrence of premature luteinization was very low in that study (6% vs. 0% in the intervention group and controls, respectively). Paradoxically, a secondary analysis of the trial by Bulow et al. [24] showed that letrozole cotreatment conversely increased serum progesterone values during ovarian stimulation compared to unexposed controls. Notably, those patients received a different ovarian stimulation protocol compared to our patients (i.e., fixed daily dose of rFSH with the addition of letrozole during the entire course of ovarian stimulation). In our study, letrozole administration lasted for five days and was combined with CFα, followed by “dose-adjusted” daily rFSH. Additionally, our patients obtained a higher mean number of oocytes compared to those in the study by Poulsen et al. [24]. All these factors may have caused different trends in progesterone growth between studies, with a general tendency towards higher mean progesterone values in our study population. In this respect, we believe that the lower progesteronemia found in Group_A was not linked to direct effects of letrozole on the progesterone biosynthetic pathway, but rather to the possibility of achieving a mild stimulation of FSH receptors during the second phase of ovarian stimulation (i.e., from the eighth day until ovulation induction) after a first phase of sustained follicle development [25]. This fact may have led to lower commitment of CYP17-A1, with reduced release of progesterone into the blood stream.

The lower estradiol and progesterone values observed in Group_A on the day of ovulation induction are particularly noteworthy, as they seem to have conferred several benefits to the patients. In addition to the higher rate of fresh embryo transfer, which may have reduced the time and costs associated with cycle segmentation, the decreased levels of progesterone may have played a crucial role in promoting optimal endometrial receptivity. Indeed, high progesterone levels have been associated with a variety of negative effects on endometrial function, including impaired receptivity and decreased implantation rates [26]. By reducing progesterone levels while still maintaining oocyte competence, the novel stimulation protocol used in Group_A may thus have contributed to improved pregnancy outcomes for these patients [27]. Moreover, the observed differences in serum estradiol levels may also have had a beneficial impact on ovarian function, as excessive estradiol levels have been linked to a range of complications, including ovarian hyperstimulation syndrome (OHSS). Taken together, these findings suggest that the use of a combination of CFα and letrozole may offer several advantages over traditional r-FSH-based protocols for IVF, including improved ovarian response and endometrial receptivity, as well as reduced risk of complications.

### 4.2. Strengths and Limitations

Originality and strict inclusion criteria were the main strengths of our study. The main limitations include the observational design of the study and the small sample size between comparators. Additionally, the variable rFSH starting dose in Group_B and possible dose adaptation during the time of stimulation in both groups were potential sources of bias, limiting the consistency of our findings.

## 5. Conclusions

The combination of CFα plus five days of letrozole yielded similar results compared to daily rFSH in terms of MII oocytes, IR, CPR, and LBR in E-NOR. Nevertheless, it was associated with a trend towards a higher number of developing follicles, total oocytes, and obtained embryos.

Additionally, CFα and letrozole co-treatment resulted in lower estradiol and progesterone values at ovulation induction compared to daily rFSH, resulting in an increased chance of performing a fresh embryo transfer with no OHSS occurrence.

Whilst these findings are encouraging, no definitive conclusion can be drawn on the superiority of the experimental protocol compared with rFSH due to the observational design of our study. Therefore, further well-conducted RCTs evaluating the combination of CFα plus letrozole in E-NOR are needed.

## Figures and Tables

**Figure 1 diagnostics-13-01249-f001:**
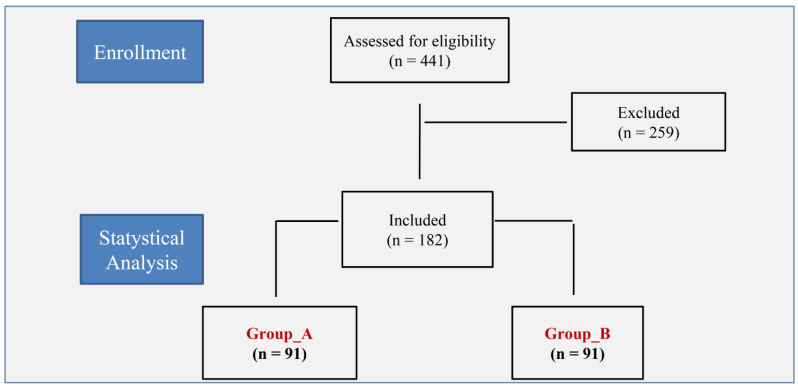
Flow diagram of the study.

**Figure 2 diagnostics-13-01249-f002:**
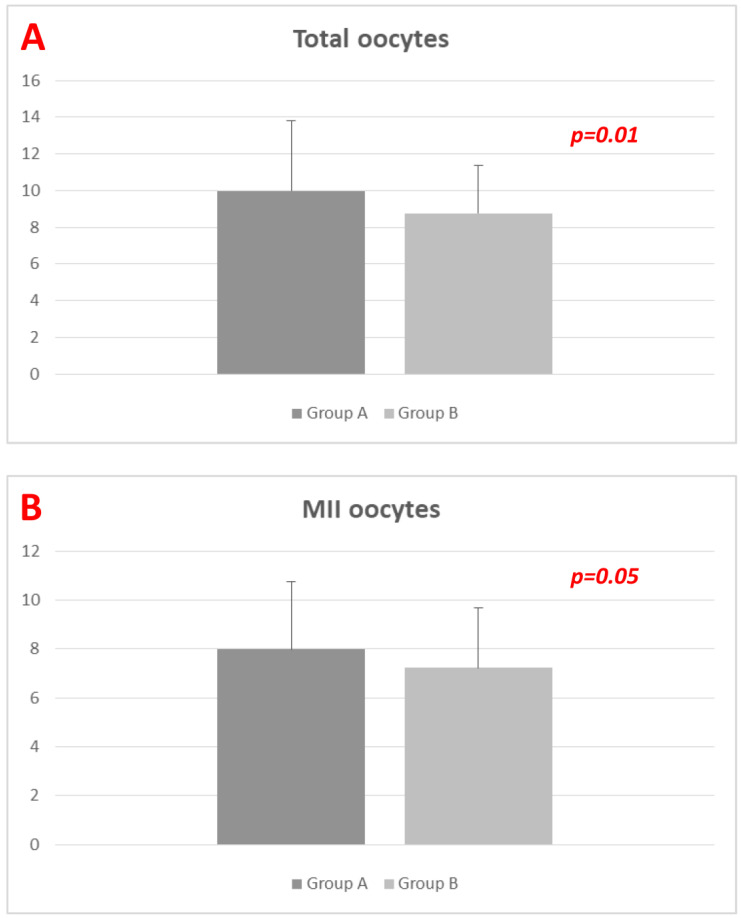
(**A**,**B**). Number of retrieved total oocytes (**A**) and MII oocytes (**B**) among groups (data expressed as mean ± standard deviation). *p*-values obtained by using Student’s *t*-test.

**Figure 3 diagnostics-13-01249-f003:**
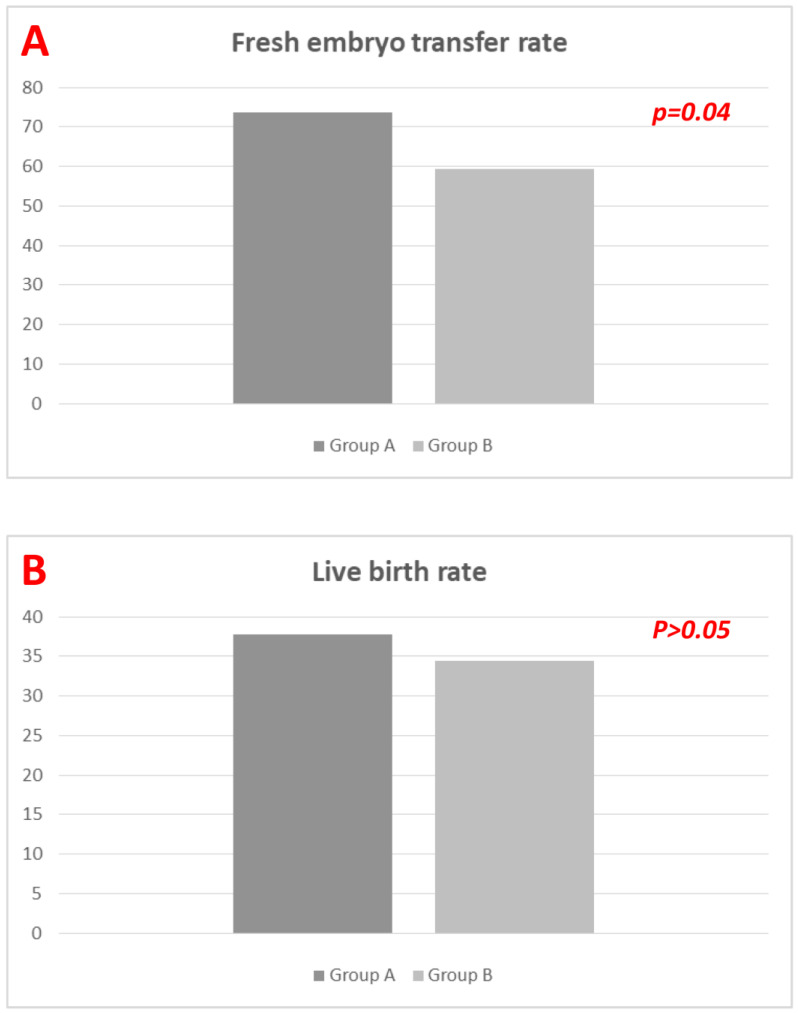
(**A**,**B**). Fresh embryo transfer rate (**A**) and Live birth rate (**B**) among groups (data expressed as percentages). *p*-values obtained by using Fisher’s *t*-test.

**Table 1 diagnostics-13-01249-t001:** General features of the study population (categoric and continuous variables expressed as number/percentage, or mean ± standard deviation, respectively).

Variables		Group_A (*n* = 91)	Group_B (*n* = 91)
Age (years)		32.42 (±2.34)	31.90 (±2.59) °
BMI (kg/h^2^)		22.3 (±3.0)	22.0 (±2.9) °
Spontaneous miscarriage		0.2 (±0.4)	0.4 (±0.3) °
Cause of infertility	Unexplained	33% (30)	38.5% (35) °
	Mild male factor	19.8% (18)	23.1% (21) °
	Tubal factor	38.5% (35)	33% (30) °
	Endometriosis	8.8% (8)	5.5% (5) °
Duration of infertility (months)		18.0 (±9.2)	21.3 (±8.5) °
AFC		11.31 (±3.16)	11.14 (±2.04) °
FSH (iU/mL)		7.08 (±1.4)	6.95 (±1.4) °
LH (iU/mL)		4.9 (±1.60)	5.8 (±2.10) *
E2 (pg/mL)		46.6 (±20.9)	42.3 (±14.5) °
AMH (ng/mL)		2.49 (±0.76)	2.78 (±0.90) °

° non-statistically significant * statistically significant.

**Table 2 diagnostics-13-01249-t002:** Outcomes of the IVF cycles among groups (categoric and contiuous variables expressed as number/percentage or mean ± standard deviation, respectively).

Variables	Group_A (*n* = 91)	Group_B (*n* = 91)
Duration of ovarian stimulation (days)	11.39 ± 1.66	10.98 ± 1.37 °
Daily r-FSH consumption (IU)	692.7 ± 487.9	2396.5 ± 1122.1 *
Long acting FSH consumption (IU)	131.32 ± 24.32	-
GnRH-antagonist injections (number)	4.06 ± 1.42	4.88 ± 1.24 *
Day 8 antral follicles (number)	9.89 ± 4.4	5.74 ± 3.87 *
Pre-ovulatory follicles (number)	8.74 ± 3.30	7.96 ± 2.61°
Serum estradiol on the day of ovulation induction (ng/mL)	1947.9 ± 1361.1	2681.2 ± 1569.2 *
Serum progesterone on the day of ovulation induction (ng/mL)	1.10 ± 0.57	1.35 ± 0.92 *
Total oocytes (number)	9.97 ± 3.81	8.76 ± 2.60 *
MII oocytes (number)	7.98 ± 2.78	7.22 ± 2.47 °
Fertilization rate (proportion)	0.79 ± 0.22	0.80 ± 0.19 °
Total embryos (number)	3.73 ± 2.19	3.07 ± 1.22 *
Good quality embryos/blastocysts (number)	2.72 ± 1.03	2.87 ± 1.19 °
Fresh transfer (%)	73.6%	59.3% *
Transferred embryos (number)	1.52 ± 0.57	1.52 ± 0.57 °
Transferred good quality embryos/blastocysts (number)	1.39 ± 0.62	1.41 ± 0.54 °
Positive HCG test (%)	53.8%	46.1% °
Implantation rate (%)	38.4%	35.5% °
Cinical pregnancy rate (%)	46.7%	43.7% °
Miscarriage rate (%)	16.7%	11.8% °
Live birth rate (%)	35.6%	38.8% °

° non-statistically significant * statistically significant.

## Data Availability

The data presented in this study are available in this manuscript.

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
