# Peer review of "Corifolitropin-Alfa plus Five Days Letrozole Versus Daily Recombinant-FSH in Expected Normo-Responder Patients: A Retrospective Comparative Study"

_diagnostics, 2023, doi:10.3390/diagnostics13071249_

Round 1

Reviewer 1 Report

Dear Authors,

Thank you very much for your very promising experimental protocol.

A few corrections required.

Line 48: The rationale.. Please define.

Line 125: Vitrification. Please analyze.

Line 150: Reference is not written to the reference list. Also, it is ont written in the proper numeral way.

In Figure 2, please change the colors of the diagrams. It would also be useful to increase the dimensions of the figure. The same mist be applied also to Figure 3.

Line 237: Please change the expression of the phrase.

Thank you very much in advance.

Author Response

RE: diagnostics-2267109

Dear Editor:

Thank you for giving us the chance to enhance our manuscript “Corifolitropin-α plus five days letrozole versus daily recombinant-FSH in expected normo-responder patients: a retrospective comparative study.”

Below are each question raised, followed by our response, as well as the position in the paper where issue is mentioned.

In the present version of the manuscript, we addressed all the issues and modified Figures 2 and 3 following the suggestions made by Reviewers.

REVIEWER #1

Dear Authors,

Thank you very much for your very promising experimental protocol.

A few corrections required.

Response: We are sincerely grateful for your words of appreciation and for giving us the possibility to improve the overall quality of our study. We now addressed all the issue raised in the revised version of the manuscript. 

Reviewer#1 Comment #1

A) Line 48: The rationale.. Please define.
B) Response: We now better clarified this point.

C) Pre-existing text Location: Lines 48-49.

D) Modified text: The reason for creating new protocols in IVF programs is to make them more affordable and efficient, and to reduce the amount of stress and difficulty involved in undergoing treatment.

Reviewer#1 Comment #2

A) Line 125: Vitrification. Please analyze.

B) Response: Thank you for this suggestion. Vitrification/warming kit and operators were now specified.

C) Pre-existing text Location: Line 125.

D) Modified text: Vitrification/warming was performed by two operators by using the kit produced by Kitazato BioPharma Co. (Japan).

Reviewer#1 Comment #3

A) Reference is not written to the reference list. Also, it is ont written in the proper numeral way.

B) Response: Thank you for your observation. The appropriate reference (and style) was now included.

Reviewer#1 Comment #4

A) In Figure 2, please change the colors of the diagrams. It would also be useful to increase the dimensions of the figure. The same mist be applied also to Figure 3.

B) Response: The size and colours of Figure 2 and 3 were now improved.

Reviewer#1 Comment #5

A) Line 237: Please change the expression of the phrase.

B) Response: Thank you. The sentence was now rephrased.

C) Modified text Location: Line 237.

D) Modified text: Otherwise, if we look at the data from another perspective, the lower rate of oocyte maturity in Group_A was potentially ascribable to asynchrony between follicle growth and inner oocyte maturation due to the changes in follicular fluid dynamics exerted by letrozole.

Thank you and we look forward to hearing from you.

Sincerely,

Amerigo Vitagliano, M.D. (for all authors).

Reviewer 2 Report

Many recent studies have evaluated the role of corifollitropin alfa or that of letrozole in ovarian stimulation. The same as, rFSH, corifollitropin alfa interacts only with the FSH receptor and has no LH activity.  Letrozole administered for ovulation induction has a more successful rate of ovulation. 

This study evaluates the effectiveness of a new ovarian stimulation protocol based on the combination of corifollitropin -alfa and five days of letrozole administration.

The oocyte pick-up was completed without complications and the above combination was associated with a tendency a higher number of developing follicles, total oocytes and obtained embryos. CF-alfa and letrozole co-treatment determined lower estradiol and progesterone values in ovulation associated with induction incomparison to the daily administration of rFSH.

The originality and strict inclusion criteria were the main strong points of this study.  The bibliography  presented included recent titles. The article is well written and documented. I recommend it for publication.

A little clarification is necessary.  A concordance between the name corifolitropin alfa used in the title and the name corifollitropin -alfa in the presented text is recommended.

Author Response

RE: diagnostics-2267109

Dear Editor:

Thank you for giving us the chance to enhance our manuscript “Corifolitropin-α plus five days letrozole versus daily recombinant-FSH in expected normo-responder patients: a retrospective comparative study.”

Below are each question raised, followed by our response, as well as the position in the paper where issue is mentioned.

In the present version of the manuscript, we addressed all the issues and modified Figures 2 and 3 following the suggestions made by Reviewers.

REVIEWER #2

Many recent studies have evaluated the role of corifollitropin alfa or that of letrozole in ovarian stimulation. The same as, rFSH, corifollitropin alfa interacts only with the FSH receptor and has no LH activity.  Letrozole administered for ovulation induction has a more successful rate of ovulation.

This study evaluates the effectiveness of a new ovarian stimulation protocol based on the combination of corifollitropin -alfa and five days of letrozole administration.

The oocyte pick-up was completed without complications and the above combination was associated with a tendency a higher number of developing follicles, total oocytes and obtained embryos. CF-alfa and letrozole co-treatment determined lower estradiol and progesterone values in ovulation associated with induction incomparison to the daily administration of rFSH.

The originality and strict inclusion criteria were the main strong points of this study.  The bibliography  presented included recent titles. The article is well written and documented. I recommend it for publication.

A little clarification is necessary.  A concordance between the name corifolitropin alfa used in the title and the name corifollitropin -alfa in the presented text is recommended.

Response: Thank you so much for enhancing our attempts to evaluate the effectiveness of a novel ovarian stimulation protocol for expected normo-responder patients. In the revised version of the manuscript, we ensure that only the term “corifollitropin-alfa” and the acronym “CFα” are mentioned.

Thank you and we look forward to hearing from you.

Sincerely,

Amerigo Vitagliano, M.D. (for all authors).